# Safety and Efficacy of Single-Fraction Carbon-Ion Radiotherapy for Early-Stage Lung Cancer with Interstitial Pneumonia

**DOI:** 10.3390/cancers16030562

**Published:** 2024-01-29

**Authors:** Shuri Aoki, Hitoshi Ishikawa, Mio Nakajima, Naoyoshi Yamamoto, Shinichiro Mori, Tokuhiko Omatsu, Yuji Tada, Teruaki Mizobuchi, Satoshi Ikeda, Ichiro Yoshino, Shigeru Yamada

**Affiliations:** 1QST Hospital, National Institutes for Quantum Science and Technology, 4-9-1 Anagawa, Inage-ku, Chiba-shi 263-8555, Japan; aoki.shuri@qst.go.jp (S.A.); nakajima.mio@qst.go.jp (M.N.); n.yamamoto@chouseihp.jp (N.Y.); mori.shinichiro@qst.go.jp (S.M.); omatsu.tokuhiko@qst.go.jp (T.O.); yamada.shigeru@qst.go.jp (S.Y.); 2Department of Pulmonary Medicine, International University of Health and Welfare, Narita Hospital, Hatakeda 852, Narita 286-8520, Japan; ytada25@yahoo.co.jp; 3Department of General Thoracic Surgery, Social Welfare Organization Saiseikai Imperial Gift Foundation, Chibaken Saiseikai Narashino Hospital, 1-8-1 Izumi-Cho, Narashino-shi 275-8580, Japan; tmizobuc@gmail.com; 4Department of Respiratory Medicine, Kanagawa Cardiovascular and Respiratory Center, 6-16-1, Tomioka-higashi, Kanazawa-ku 236-0051, Japan; ikeda.0880f@kanagawa-pho.jp; 5Department of Thoracic Surgery, International University of Health and Welfare, Narita Hospital, Hatakeda 852, Narita 286-8520, Japan; iyoshino@iuhw.ac.jp

**Keywords:** carbon-ion radiotherapy, lung cancer, interstitial pneumonia, radiation pneumonitis

## Abstract

**Simple Summary:**

Patients with lung cancer complicated by interstitial pneumonia (IP) often lose treatment options early because of concerns about acute exacerbation of IP. Carbon-ion radiotherapy (CIRT) is expected to provide both superior tumor control and low toxicity owing to its superior dose concentration; however, it is not yet a well-established therapy. In this study, we confirmed that 50 Gy single-fraction CIRT can be performed even in IP-complicated lung cancer with acceptable efficacy and tolerability. Lung dose is a significant predictor of overall survival, indicating the need for further efforts to reduce lung dose.

**Abstract:**

Patients with lung cancer complicated by interstitial pneumonia (IP) often lose treatment options early owing to acute exacerbation of IP concerns. Carbon-ion radiotherapy (CIRT) can provide superior tumor control and low toxicity at high dose concentrations. We conducted a retrospective analysis of the efficacy and tolerability of a single-fraction CIRT using 50 Gy for IP-complicated lung cancer. The study included 50 consecutive patients treated between April 2013 and September 2022, whose clinical stage of lung cancer (UICC 7th edition) was 1A:1B:2A:2B = 32:13:4:1. Of these, 32 (64%) showed usual interstitial pneumonia patterns. With a median follow-up of 23.5 months, the 3-year overall survival (OS), cause-specific survival, and local control rates were 45.0, 75.4, and 77.8%, respectively. The median lung V5 and V20 were 10.0 and 5.2%, respectively (mean lung dose, 2.6 Gy). The lung dose, especially lung V20, showed a strong association with OS (*p* = 0.0012). Grade ≥ 2 pneumonia was present in six patients (13%), including two (4%) with suspected grade 5. CIRT can provide a relatively safe and curative treatment for patients with IP-complicated lung cancer. However, IP increases the risk of severe radiation pneumonitis, and further studies are required to assess the appropriate indications.

## 1. Introduction

Lung cancer (LC) is often associated with interstitial pneumonia (IP). Up to 20% of patients with idiopathic pulmonary fibrosis (IPF), the most common form of IP, are reported to develop LC, with an estimated five-fold incidence [1,2,3]. However, treatment options for IP-complicated LC are limited, even in the earliest stages, because of concerns that all treatment modalities, such as surgery, chemoimmunotherapy, and radiation therapy (RT), can induce acute exacerbations of IPF [4,5,6,7,8,9,10,11,12]. RT for early-stage LC, particularly stereotactic body RT (SBRT) for non-small cell LC (NSCLC), has proven to be an effective and well-tolerated treatment for patients who are medically inoperable. However, most clinical trials have excluded cases of pre-existing IP, making reaching a consensus on the safety of SBRT in such patients challenging. The risk factors and their borderlines remain unclear, and treatment for such patients has not improved.

Conversely, particle therapy has a favorable dose concentration among radiation modalities [13,14], and when treating lung tumors, the dose for normal lungs could be minimized. Therefore, even for IP-complicated LC, which is difficult to treat with other modalities, curative treatment can be provided with relative safety [15,16]. In particular, carbon-ion radiation therapy (CIRT) exhibits biological effects because of its high linear energy transfer radiation [17]. The National Institute of Radiological Sciences, currently the QST Hospital, initiated CIRT for NSCLC in 1994. Based on the results of dose-escalation studies, a single irradiation protocol of 50 Gy (described as the relative biological effect [RBE]-weighted dose based on the modified microdosimetric kinetic model) has been employed for early-stage LC [18] and has achieved extremely high safety and efficacy in cases without IP [19,20,21]. However, even for CIRT, limited clinical data on its safety are available for patients with IP complications. In this study, we evaluated the toxicity and efficacy of single-irradiation CIRT for NSCLC complicated by IP, especially analyzing the risk factors for acute exacerbation of IP.

## 2. Materials and Methods

### 2.1. Study Design and Patient Selection

Consecutive patients treated with CIRT for NSCLC complicated by IP between April 2013 and September 2022 were identified and retrospectively analyzed. The treatment protocols and procedures were approved by our institution’s ethics committee, and written informed consent was obtained from all patients who participated in the study. The inclusion criteria for the study were as follows: (1) histologically or pathologically confirmed NSCLC treated with single-fraction CIRT with 50 Gy, (2) clinical stage 1–2 (UICC7th) with no nodal or distant metastasis, (3) medically inoperable or refused surgery, (4) Eastern Cooperative Oncology Group Performance Status (PS) 0–2, and (5) patients who met the above criteria and were clinically or pathologically diagnosed with IP. Central tumors were not included in this analysis, as they were treated using a different protocol with 12 fractions due to safety concerns. The availability of treatment with other modalities (surgery, photon SBRT), acceptability of CIRT, and treatment strategies were determined by a multidisciplinary cancer board, which included thoracic surgeons and physicians. Screening and diagnosis of IP were conducted at referral institutions prior to or in parallel with the diagnosis of LC and were also reviewed at our institution based on clinical and laboratory findings. The usual interstitial pneumonia (UIP) pattern was diagnosed using high-resolution computed tomography (CT) images, according to the new international guidelines for IPF diagnosis [22]. Two radiation oncologists and one radiologist made the final diagnosis. The GAP model (and the revised GAP model used in Japan; rGAP model) for predicting the prognosis of IPF [23,24] was also used to evaluate the patients.

### 2.2. Carbon-Ion Radiotherapy Procedure

The details of the CIRT planning and delivery at our institution have been described previously [18,19,25,26,27,28]. The carbon ion dose was calculated by multiplying the physical dose by the RBE and was expressed in Gy. A fixed dose of 50 Gy was delivered in a single fraction through a series of 2–4 fixed ports. Treatment planning for CIRT was conducted using four-dimensional (4D) CT at 1–2-mm intervals. The gross tumor volume (GTV) was delineated, including the lung tumor at the lung window. The clinical target volume (CTV) was created by adding a margin of 0.5–1.0 cm to the GTV. In addition, we defined the beam field-specific target volume (FTV) by extending the 3D treatment planning technique to 4D [29] and a setup margin of 2–3 mm to create the planning target volume (PTV). The total dose was applied to the isocenter and tuned to cover the PTV with a 95% isodose line of the prescribed dose. For organs at risk (OARs), dose constraints were set based on our previous clinical trials [18] and strictly adhered to the following criteria: spine (Dmax) < 10 Gy, esophagus (D0.2cc) < 10 Gy, and mainstem bronchus (D2cc) < 30 Gy, taking priority over the target coverage. No dose constraints on the lung doses were defined.

### 2.3. Follow-Up

Post-CIRT follow-up consisted of physical examination, blood tests, chest radiography, and contrast-enhanced CT performed every 3 months for the first 2 years and thereafter, at least every 6 months. In cases where continuous examinations at our hospital were difficult, the latest medical reports and diagnostic images were sent to us. Local recurrence was defined as a progressive abnormality on CT. In cases of suspected local recurrence, 18F-fluorodeoxyglucose positron emission tomography was performed, and when feasible, a biopsy was performed. Acute and late radiation pneumonitis (RP) was assessed using the Common Terminology Criteria for Adverse Events (CTCAE) Version 5.0 [29]. Acute AEs were defined as AEs occurring within 3 months and late AEs as those occurring later. Many patients had respiratory symptoms before CIRT; therefore, their symptoms were graded before and after CIRT, and the onset or progression of symptoms after treatment was evaluated. Mild progression (addition or change of medication due to exacerbation) was classified as grade 2, and severe exacerbation (need for inpatient treatment and introduction of home oxygen therapy [HOT]) was classified as grade 3. Grades 4 and 5 were the same as those used in the common evaluation.

### 2.4. Chart Review

The doses to the target and OARs were evaluated using dose-volume histogram (DVH) analysis. In this study, as indicators of the dose to normal lungs (lungs-GTV), the mean total lung dose (MLD total) and percentage of lung volume irradiated above 5/10/20 Gy (lungs V5/V10/V20) were retrieved. Overall survival (OS), cancer-specific survival (CSS), local control, and progression-free survival (PFS) were analyzed using the Kaplan–Meier method, and subgroups were compared using log-rank statistics. Cox proportional hazards regression methods were used for risk factor analysis. Fisher’s exact test was used to examine the independence of events and risk factors. Each outcome was calculated from the date of CIRT initiation to the date of the event or the last follow-up date. Statistical analyses were performed using R software version 4.2.3 (https://www.r-project.org/, accessed on 23 December 2023), and statistical significance was set at *p* < 0.05. Each indicator of respiratory function and serum marker was evaluated based on borderline values.

### 2.5. Patient Characteristics

We identified patients with NSCLC who received single-fraction CIRT at 50 Gy between April 2013 and September 2022. Among these, 50 cases complicated by IP were included in this study. Patient characteristics are summarized in Table 1. The median age of the patients was 76 (interquartile range [IQR] 71–81) years, and the male-to-female ratio was 45:5. The clinical stage was 1A:1B:2A:2B = 32:13:4:1. All participants were clinically or pathologically diagnosed with IP. Of these, 32 (64%) showed UIP patterns and were diagnosed with IPF, and 9 (18%) were suspected to be associated with autoimmune diseases (mainly rheumatoid arthritis, other dermatomyositis, and anti-neutrophil cytoplasmic antibody: ANCA-related diseases). Eleven (22%) and five (10%) patients were on corticosteroids and HOT, respectively, and seven (14%) had a history of acute exacerbations. The proportions of patients with GAP stage 1:2:3 and rGAP stage 1:2:3 were 26:19:5 and 22:22:6, respectively.

## 3. Results

### 3.1. Treatment Outcomes

The median follow-up time was 23.5 (IQR 10–47.5) months for all patients and 28 (IQR 9–47) months for survivors. At the final follow-up, 35 patients (70%) had died, with 12 of those deaths attributed to other causes. Of the 23 deaths from other causes, 2 were related to AEs, 13 were respiratory-related (pneumonia, respiratory failure) deaths excluding those related to AEs, and 8 were due to other causes (2 cardiac diseases, 2 other cancers, 1 gastric ulcer, and 3 unknown).

The median OS was 34 months, and the 2-, 3-, and 5-year OS rates were 60.8% (95% confidence interval [CI]: 44.5–73.7%), 45.0% (95% CI: 29.3–59.4%), and 18.9% (95%CI: 7.5–34.1%), respectively (Figure 1A). Moreover, the 2-, 3-, and 5-year CSS rates were 83.2% (95%CI: 65.9–92.2%), 75.4% (95% CI: 55.9–87.2%), 62.5% (95%CI: 38.3–79.5%), respectively (Figure 1B).

Twenty patients (40%) experienced recurrences, and the first site of recurrence was local in 7 patients, regional in 11, and distant in 11 (nine patients had two simultaneously). The 2- and 3/5-year local control rates were 82.7% (95% CI: 65.1–92.0%) and 77.8% (95% CI: 57.9–89.2%), respectively (Figure 1C). The median PFS was 22 months, and the 2- and 3/5-year PFS rates were 48.1% (95% CI: 33.1–61.7%) and 14.0% (95% CI: 49.0–27.8%), respectively (Figure 1D).

DVH data were evaluated for all patients. Table 2 presents the treatment characteristics of the patients. The lung dose was overestimated in four patients because part of the lung was out of range for the planning CT. The median GTV volume was 7.15 (IQR 3.80–15.08) cm^3^. In all cases, the GTV dose was within the prescribed range of ±5%. The doses delivered to the main OARs are listed in Table 2. For the lungs, the lung V5, V10, and V20 were 10.0 (IQR 6.49–12.12)%, 7.59 (IQR 5.23–10.23)% and 5.18 (IQR 3.72–6.92)%, respectively. The MLD was 2.63 (IQR 1.87–3.18) Gy.

### 3.2. Prognostic Factors for OS

Table 3 summarizes the results of the univariate analysis of the factors associated with OS. Factors known to be associated with OS in early-stage LC, including age, PS, and cancer stage, were not significantly predictive of OS in this study. In addition, the parameters reported to be useful in predicting the prognosis of interstitial pneumonia, such as causative diseases, serum markers, and respiratory function, were not significantly associated with OS. In contrast, regarding dosimetric variables, V5, V20, and MLD showed significant associations in univariate analysis, and multivariate analysis indicated that V20 remained a significant risk factor. (*p* = 0.0012). The association between GAP/rGAP stage and OS was not significant (*p* = 0.28/0.58).

### 3.3. Adverse Events

AEs were discussed in 49 of the 50 patients, excluding one patient who died of the primary disease within 2 months. Of the 49 patients in the present study, RP of grades 2, 3, and 5 were encountered in two patients (4.1%) each, totaling six patients (13%) (Table 4). Figure 2 shows the pretreatment thin-slice CT and dose distribution of the two patients with grade 5 pneumonia. The median duration of onset was 3 months (range, 1–6 months). Two patients who developed grade 5 RP presented with symptoms within 1 month of starting CIRT, indicating that early onset should be carefully monitored as a sign of severe pneumonia. Both patients died of respiratory failure 5 months after treatment and multiple steroid pulse therapies. No tumor recurrence was observed in either of the patients. Five of 32 patients (15.6%) with a UIP pattern developed grade 2 or higher RP, whereas only one (5.9%) of 17 patients without a UIP pattern. Statistical evaluation failed to show any significant correlation between RP and any of the patient backgrounds, serum markers, respiratory function, or lung irradiation doses previously reported to be associated with RP. When patients were classified using the rGAP model, RP tended to be present in patients with a higher rGAP stage (Table 5).

## 4. Discussion

### 4.1. Radiation Therapy for IP-Complicated LC

This retrospective study evaluated the efficacy and safety of a single CIRT in 50 patients with IP-complicated localized LC. This is one of the largest reports on radical irradiation in patients with IP and, to our knowledge, the first report of single-carbon-beam irradiation.

In photon radiotherapy, ILD complications have long been identified as risk factors for severe RP [30,31,32,33,34,35]. In particular, patients diagnosed with IP are at a high risk of fatal RP and are considered unsuitable for irradiation, resulting in limited reports [10,36,37,38]. Yamashita et al. reported that seven of nine patients with IP (78%) who underwent SBRT with 48 Gy in four fractions developed grade 4–5 pneumonia [10]. Tsurugai et al. reported that SBRT for 42 idiopathic interstitial pneumonias complicated by LCs resulted in grade ≥ 3 RP in 12% of patients [36]. These results indicate that the risk of fatal RP is considerably higher in patients with IP complications, even with SBRT for stage I NSCLC.

In contrast, particle therapy, that is, proton beam therapy (PBT) and CIRT, has been increasingly reported to have a relatively low risk of irradiation in cases with pulmonary interstitial changes [26,39,40,41,42,43]. Among them, studies focusing on IP are still scarce; however, Hashimoto et al. reported in 2019 that 29 patients with IP-complicated LC (including 10 patients with UIP pattern) underwent PBT, resulting in four patients (13.8%) with grade 2–3 RP [40]. Noh JM et al. reported in 2020 the results of radical PBT in 54 patients with IP-complicated LC, resulting in seven patients (13.0%) of grade ≥ 3 pneumonia [41]. The present study found grade ≥ 2 RP in 6 of 50 patients (12%). This is comparable to previous reports on particle therapy for IP-complicated LC, suggesting that our single CIRT is not inferior in safety (Table 6) [10,36,40,41]. In contrast, in our previous report on the same regimen for early-stage LC without IP, none of the 57 patients had G2 or higher pneumonitis [19].

### 4.2. Patient Background Related to Severe RP

The UIP pattern on high-resolution computed tomography (HRCT) is a well-known risk factor for AEs in the natural history of IPF [44,45] and an important predictor of pneumonitis due to cancer treatment, including radiotherapy [38,46,47]. In this study, the incidence of grade ≥ 2 pneumonia was approximately 2.7 times higher in the group with UIP pattern than in those without it (16% in UIP vs. 6% in non-UIP). This supports previous reports on the association between the UIP pattern and IP severity. Furthermore, the only statistically significant risk factor for pneumonia in this study was pretreatment transcutaneous oxygen saturation (SpO_2_ < 95%). This is a brief indicator of the arterial partial pressure of oxygen (PaO_2_ < 80%) used in the Japanese IPF severity classification [48]. This is consistent with previous reports and the clinical sense that higher IPF severity is associated with a higher risk of severe pneumonia [35,49,50,51].

Other reported parameters include patient background, such as age and sex, collagen disease, IP history (corticosteroids, HOT, and AEs), and several respiratory function indicators [33,52,53]. Serum markers have been studied, particularly in the field of radiotherapy, with KL-6, SP-D, and CRP levels being the most common [10,27,54,55]. These indicators were not significantly associated with the severity of pneumonia in the present study. This could be partly due to bias and the small sample size; however, it may also be a result of the complexity of the multiple factors involved in the development of pneumonia. Therefore, attempts have been made to score multiple risk factors for each treatment modality. The GAP/rGAP model evaluated in this study tended to be related to OS. With a larger number of cases, estimating the risk of radiotherapy by modifying or combining existing scores for other treatment modalities is possible.

### 4.3. Importance of Lung Doses

Previous reports have often identified lung dose (V2-25, MLD) as a risk factor for severe pneumonia [31,55,56,57]. Furthermore, the two references on PBT cited in the previous section reported inferior OS in patients with a UIP pattern/IPF diagnosis. In this study, lung doses were not observed to be significantly associated with severe pneumonia; however, lungV20 (<5%) was strongly associated with OS, and other lung dose parameters tended to be similar. One reason for this may be that the lung dose is a comprehensive indicator of tumor size, shape, and location. At the same time, radiation-induced lung fibrosis and loss of respiratory function potentially affect the long-term prognosis of patients with IP, even if they do not lead to severe pneumonia or AEs. In other words, in radiotherapy, especially in cases with IP complications, the reduction of lung doses is an important issue not only for controlling AEs but also for the prognosis and quality of life. Particle therapy, especially CIRT, has excellent depth-dose characteristics known as Bragg peaks. It allows the creation of a region called the spread-out Bragg peak (SOBP) adapted to the depth and shape of the treatment target with only several beam angles, minimizing dose scattering to the nearby OAR and increasing the dose to the target [13]. As a result, CIRT enables reduced lung doses, particularly in the low-dose area, which is probably related to lower toxicity in CIRT for IP-complicated LCs [10,32,33,34,37]. Clearly, IP complications are associated with severe pneumonia, even in CIRT, compared with the results using the same regimen for non-IP cases [19]. In addition, two cases of grade 5 RP occurred in the present study, but the underlying risk remains to be clarified. In contrast, 21 patients who met both the diagnosis of rGAP-stage 1 and treatment with lung doses lower than the set criteria (V5 < 10%, V20 < 5%, and MLD < 3 Gy), including those showing a UIP pattern, experienced no grade ≥ 2 pneumonitis. To determine the indications for irradiation in IP cases, we should consider the background factors and dose constraints from both sides or in combination. Although not directly applicable to photon SBRT owing to differences in biological effects and the number of fractions, experience with CIRT could be applied to radiotherapy in general by indicating relatively safe subjects and dose criteria for IP-complicated LC.

### 4.4. Treatment Effect

In radiotherapy for ILD/IP-complicated LC, IP itself has a significant impact on prognosis [10,32,33,34,38,58]. In the present study, mortality from other causes was higher than that from cancer. The 3-year OS was 45%, which was significantly inferior to our results with the same protocol for non-IP cases [19]. This is a common trend in reports of radiotherapy for IP-complicated LC. Among Japanese multicenter retrospective studies, the study on SBRT reported a 3-year OS of 42.6% and that on CIRT 48.2% [33,39]. Other studies have similarly reported relatively poor OS after treatment of IP-complicated LC, regardless of the radiation modality [38,40,41]. Meanwhile, it is clear from several reports that local control of radiotherapy is also reduced in IP-complicated cases [26,40,42,43]. The underlying mechanisms remain unclear, but it has been reported that chronic inflammation and fibrosis could be related to poor tumor response [59]. Also, the difficulty in distinguishing the tumor from the surrounding interstitial shadows could have led to inaccurate targeting and post-treatment assessment, or efforts to reduce lung irradiation might have been negatively reflected. In patients with IP complications, many factors should be fully considered, including the prognosis of IP, the negative effects of the treatment, and the patient’s preferences; then, a careful assessment should be made of the feasibility of the treatment intervention.

### 4.5. Limitation

This study had some limitations. First, the study was conducted at a single institution and had a retrospective design. Therefore, some bias may have been inherent. In addition, CIRT is not an accessible treatment due to its ubiquity and high treatment costs. Second, patients with IP are at a higher risk of undergoing biopsy for LC diagnosis, and in the majority of cases, clinical rather than pathological diagnosis is used. Interstitial pneumonia was also based on clinical diagnosis and was incompletely classified. Therefore, all cases were discussed with our cancer board regarding treatment indications and methods before proceeding with the treatment. Third, the small number of patients may have limited the statistical reliability of the results. Fourth, the safety of single irradiation, which we use in our hospital, has not been established for other modalities at present, making pure inter-modality comparisons difficult. Also, methods to compare the biological effects of CIRT and photon SBRT have not yet been established. Further studies with large sample sizes and basic research are needed to clarify the risk factors for severe RP and the role of CIRT in IP-complicated LCs.

## 5. Conclusions

We report our experience with a 50 Gy single CIRT for localized IP-complicated LC. CIRT has the potential to provide a relatively safe and curative treatment for patients who were previously considered difficult to treat with RT. However, IP complications increase the risk of serious RP, as reported in the present study, and are certainly high-risk even with CIRT. Further research is needed to identify the most beneficial targets and limitations of this treatment.

## Figures and Tables

**Figure 1 cancers-16-00562-f001:**
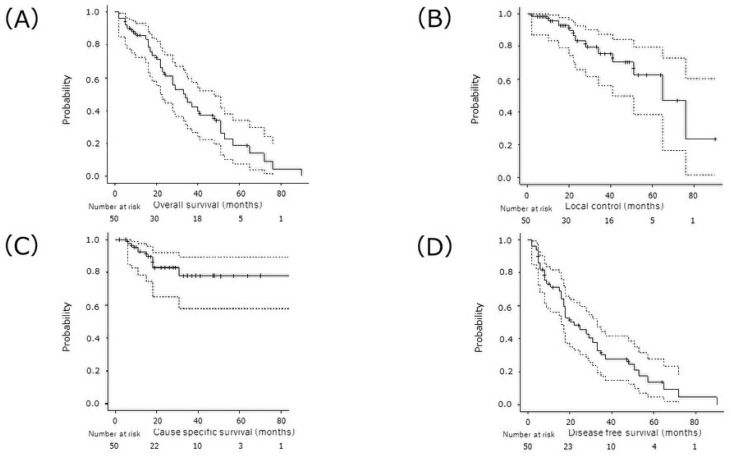
(**A**) overall survival, (**B**) local control, (**C**) cause-specific survival, and (**D**) progression-free survival rates after CIRT. CIRT, carbon-ion radiation therapy.

**Figure 2 cancers-16-00562-f002:**
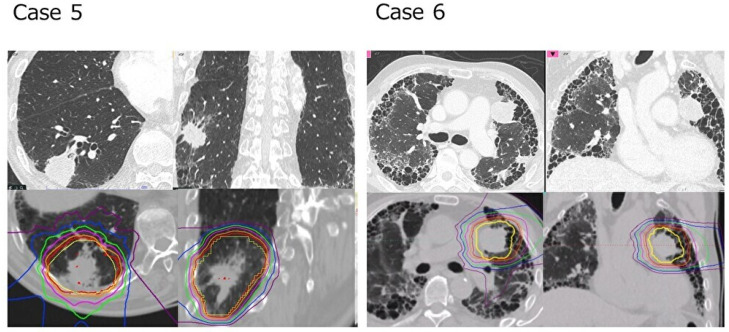
Pretreatment CT images and CIRT dose distribution of two patients with grade 5 radiation pneumonitis (cases 5 and 6 in Table 4) are shown. Both patients had a UIP pattern and clinical stage 1B (tumor diameter 39 and 31 mm, respectively). Case 5 had a mild IP, rGAP-stage 1, but a relatively higher lung dose (lungV5/V20 of 19.18%/8.07%, respectively). Case 6, on the other hand, was a patient with severe IP in rGAP-stage 3 and on HOT but with a lower lung dose (lungV5/V20 8.64%/5.48%, respectively). The red, orange, magenta, green, blue, and purple lines show dose levels of 95, 90, 70, 50, 30, and 10%, respectively; CTV is drawn in yellow.

**Table 1 cancers-16-00562-t001:** Patient Characteristics (n = 50).

Factors	Median (IQR)
Follow-up period (months)	23.5 (10–47.5)
Age (years)	76 (71–81)
Gender (female: male)	45:5
ECOG-PS (0:1:2-)	36:14:0
Brinkman index	900 (552–1245)
Pre-RT Laboratory Data	
serum KL-6 (U/L)	753 (524–858)
serum SP-D (ng/L)	109 (77–179.5)
serum CRP (mg/L)	0.20 (0.1–0.64)
Pre-RT Respiratory function	
%VC (%)	87.9 (71.1–100.2)
%FVC (%)	78.6 (64.5–90.2)
%DLco (%)	58.1 (46.1–74.5)
Operable (Yes:No)	8:42
SBRT feasibility (Yes:No)	0:50
UIP pattern (Yes No)	32:18
Autoimmune disease (Yes:No)	9:41
History of AEs (Yes:No)	7:43
Use of HOT (Yes:No)	5:45
Oral corticosteroid therapy (Yes:No)	11:39
Clinical stage (UICC7th)	
1A	32
1B	13
2A	4
2B	1
Tumor location	
Upper/Middle lobe	22
Lower lobe	28
Histology	
Adenocarcinoma	7
Squamous cell carcinoma	10
Clinically diagnosed	33

Abbreviations: IQR, interquartile range; ECOG, Eastern Cooperative Oncology Group; PS, performance status; RT radiation therapy; KL-6, Krebs von den Lungen-6; SP-D, pulmonary surfactant protein D; CRP, C-reactive protein; VC, vital capacity; FVC, forced vital capacity; DLco, diffusing capacity of the lung carbon monoxide; SBRT, stereotactic body radiotherapy; UIP, usual interstitial pneumonia; AE, acute exacerbation; HOT, home oxygen therapy.

**Table 2 cancers-16-00562-t002:** Treatment characteristics.

Characteristics	Median	IQR
GTV		
Volume (mL)	7.15	(3.80–15.08)
Max dose	50.63	(50.43–51.18)
Min dose	49.03	(48.85–49.34)
Lung-GTV		
MLD (Gy)	2.63	(1.87–3.18)
V5Gy (%)	10.00	(6.49–12.12)
V10Gy (%)	7.59	(5.23–10.23)
V20Gy (%)	5.18	(3.72–6.92)

Abbreviations: IQR, interquartile range; GTV, gross tumor volume; MLD, mean lung dose; V5/10/20 Gy, percentage of lung volume irradiated above 5/10/20 Gy.

**Table 3 cancers-16-00562-t003:** Patients and treatment factors associated with OS.

		Univariate	Multivariate
Variable		*p*-Value	*p*-Value
Age	>75 vs. ≤75	0.32	
Gender	male vs. female	0.92	
T stage (UICC8th)	1 vs. 2	0.43	
SpO_2_	<90 vs. ≥90	0.60	
History of AEs	Yes vs. No	0.92	
PSL	Yes vs. No	0.75	
Autoimmune disease	Yes vs. No	0.44	
UIP	Yes vs. No	0.91	
Respiratory function			
%VC	<80 vs. ≥80	0.70	
%DLco	<65 vs. ≥65	0.92	
%FEV1	<70 vs. ≥70	0.84	
pre-CIRT serum markers			
KL-6	>500 vs. ≤500	0.22	
SP-D	>110 vs. ≤110	0.53	
CRP	>0.3 vs. ≤0.3	0.39	
DVH parameter			
lungV5 (%)	>10 vs. ≤10	0.030 *	
lungV20 (%)	>5 vs. ≤5	0.00050 *	0.0012 *
mean lung dose (Gy)	>3 vs. ≤3	0.027 *	

Abbreviations: SpO_2_, saturation of percutaneous oxygen; AE, acute exacerbation; PSL, prednisolone; UIP, usual interstitial pneumonia; VC, vital capacity; DLco, diffusing capacity of the lung carbon monoxide; FEV1, forced expiratory volume in 1 s; CIRT, carbon-ion radiation therapy; KL-6, Krebs von den Lungen; SP-D, pulmonary surfactant protein D; CRP, C-reactive protein; DVH, dose-volume histogram; lungV5/20 Gy, percentage of lung volume irradiated above 5/20 Gy. *, *p* < 0.05.

**Table 4 cancers-16-00562-t004:** Pre-irradiation conditions and characteristics of six patients with G2-5 RP.

Case No.	RP Grade	Onset Time	c-Stage	Location	BI	rGAP Stage	Treatment	KL-6	SP-D	CRP	UIP Pattern	V5 (%)	V20 (%)	MLD (Gy)
1	2	1.5M	1A2	Rt-U	1260	2		638	200	1.02	+	6.46	3.88	2.04
2	2	4M	1A3	Rt-L	900	3	HOT, PSL	805	65.6	0.12	+	12.87	7.21	3.27
3	3	6M	1A3	Lt-U	900	2		1695	150	0.82	+	11.55	7.33	3.37
4	3	3M	1A2	Lt-L	800	2		388	52.9	0.07	-	6.68	3.88	1.9
5	5	1M	1B	Rt-L	900	1		544	105	0.75	+	19.18	8.07	4.22
6	5	1M	1B	Lt-U	680	3	HOT, PSL	824	106	0.59	+	8.64	5.48	2.75

Abbreviations: RP, radiation pneumonitis; BI; brinkman index; KL-6, Krebs von den lungen 6; SP-D, Pulmonary Surfactant Protein D; CRP, C-reactive protein; UIP, usual interstitial pneumonia; V5/20, percentage of lung volume irradiated above 5/20 Gy; MLD, mean lung dose; M, month; HOT, home oxygen therapy; PSL, prednisolone.

**Table 5 cancers-16-00562-t005:** Relationship between G2-5 RP and pre-CIRT factors.

Pre-SBRT Factors		G2-5 RP	G0-1 RP	Total	*p* Value *
Age	≥75	4	25	29	1
	<75	2	18	20	
Gender	Male	5	39	44	0.50
	Female	1	4	5	
Smoking history	BI ≥ 1000	1	20	21	0.22
	BI < 1000	5	23	28	
UIP pattern in CT	(+)	5	26	31	0.39
(−)	(−)	1	17	18	
History of AEs	(+)	1	6	7	1.00
	(−)	5	37	42	
Severity of IP	SpO_2_ < 95%	4	10	13	0.048 *
	SpO_2_ ≥ 95%	2	33	36	
Respiratory function					
%VC	<80%	2	17	19	1.00
	≥80%	4	26	30	
%DLco	<65%	5	25	30	0.38
	≥65%	1	18	19	
%FEV1	<70%	5	25	30	0.38
	≥70%	1	18	19	
Serum markers					
Serum KL-6	≥500 U/mL	5	36	41	1.00
	<500 U/mL	1	7	8	
Serum SP-D	≥110 ng/mL	2	20	22	0.67
	<110 ng/mL	4	23	27	
Serum CRP	≥0.3 mg/dL	4	18	22	0.39
	<0.3 mg/dL	2	25	27	
DVH parameters					
lung V5	>10%	3	22	25	1.00
	≤10%	3	21	24	
lung V20	>5%	4	22	26	0.67
	≤5%	2	21	23	
MLD	>3Gy	3	15	18	0.66
	≤3Gy	3	28	31	
rGAP model	Stage 1	1	20	21	
	Stage 2	3	19	22	
	Stage 3	2	4	6	

* Fisher’s exact test. Abbreviations: RP, radiation pneumonitis; CIRT, carbon ion radiation therapy; SBRT, stereotactic body radiotherapy; BI, Brinkman Index; UIP, usual interstitial pneumonia; AE, acute exacerbation; IP; idiopathic interstitial pneumonias; SpO_2_, saturation of percutaneous oxygen; VC, vital capacity; DLco, diffusing capacity of the lung carbon monoxide; FEV1, forced expiratory volume in 1 s; KL-6, Krebs von den Lungen; SP-D, pulmonary surfactant protein D; CRP, C-reactive protein; DVH, dose volume histogram; lungV5/20Gy, percentage of lung volume irradiated above 5/20 Gy; MLD, mean lung dose.

**Table 6 cancers-16-00562-t006:** Previous studies on RP in patients with IP.

Auther	Year	Patients	Diagnosis	Modality	Total Dose	Fraction	Follow	OS2y	RP Grade (%)		
		(n)			(Gy)		(M)	(%)	Grade2	Grade3	Grade4–5
Yamashita	2010	13	IP	SBRT	48	4	14.7	N/A	N/A	N/A	54
Tsurugai	2017	42	IIPs	SBRT	40–60	8–22	32.4	42.2	N/A	12	0
Hashimoto	2019	29	IP	Proton	66–74	10–37	21.1	45	6.9	6.9	0
Noh	2020	15	IPF	Proton	60–69	4–30	19.8	43.9	20	20	13.3
Present study		49	IP	Carbon	50	1	23.5	60.8	4.1	4.1	4.1

Abbreviations: OS, overall survival; RP, radiation pneumonitis; SBRT, stereotactic body radiotherapy; RBE, relative biological effectiveness; M, month; interstitial pneumonia, IP; idiopathic interstitial pneumonias, IIPs; N/A, not applicable.

## Data Availability

The data presented in this study are available on request from the corresponding author.

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
