# Peer review of "Safety and Efficacy of Single-Fraction Carbon-Ion Radiotherapy for Early-Stage Lung Cancer with Interstitial Pneumonia"

_cancers, 2024, doi:10.3390/cancers16030562_

Round 1

Reviewer 1 Report

Comments and Suggestions for Authors

General comments,

The authors described early-stage lung cancer with interstitial pneumonia treated with carbon-ion radiotherapy. This study included 50 patients, and 3-year overall survival was 45.0%. Grade 2 or more radiation pneumonia were 6 patients and Grade 5 were 2 patients. The authors concluded that carbon-ion radiotherapy can be a relatively safe and curative treatment for lung cancer with interstitial pneumonia. This manuscript included important findings for the field of radiation oncology, because it is very difficult to perform radiotherapy for patients with interstitial pneumonia. However, several modifications are required to improve the manuscript.

Specific comments,

・Page 3, line 108-110. The authors described “Figure 1. (A) overall survival, (B) cause-specific survival, (C) local control, and (D) progression-free 108 survival rates by CIRT and SBRT after radiation therapy.” I considered this content was wrong. Furthermore, Figure 1 was not described.

・Page 3, Line 149. Could you explain the detail of autoimmune disease?

・Page 4, Line 154. Table 1. “Autoimmune disease” and “History of AEs” were duplicated.

・Page 6, Line 208. Title of Table 3 was not described.

・Page 7, Line 234-240. Figure 2 was not described.

Author Response

Response to the Review1 Comments

Author's Reply to the Review Report (Reviewer 1)

General comments,

The authors described early-stage lung cancer with interstitial pneumonia treated with carbon-ion radiotherapy. This study included 50 patients, and 3-year overall survival was 45.0%. Grade 2 or more radiation pneumonia were 6 patients and Grade 5 were 2 patients. The authors concluded that carbon-ion radiotherapy can be a relatively safe and curative treatment for lung cancer with interstitial pneumonia. This manuscript included important findings for the field of radiation oncology, because it is very difficult to perform radiotherapy for patients with interstitial pneumonia. However, several modifications are required to improve the manuscript.

 We are very sorry for many deficiencies in the manuscript.

Thanks to the reviewers, we were able to make some improvements to the content.

Thank you very much for your careful and kind review and comments.

Specific comments,

・Page 3, line 108-110. The authors described “Figure 1. (A) overall survival, (B) cause-specific survival, (C) local control, and (D) progression-free 108 survival rates by CIRT and SBRT after radiation therapy.” I considered this content was wrong. Furthermore, Figure 1 was not described.

Figure 1, which was sent as a separate file, has been added and errors in the inserted sections and explanatory text have been corrected. We are very sorry for these elementary errors.

・Page 3, Line 149. Could you explain the detail of autoimmune disease?

We added details on collagen diseases. Rheumatism, dermatomyositis and anti-neutrophil cytoplasmic antibody: ANCA-related diseases were included.

・Page 4, Line 154. Table 1. “Autoimmune disease” and “History of AEs” were duplicated.

Thank you for pointing this out. The upper columns were incorrect and have been removed.

・Page 6, Line 208. Title of Table 3 was not described.

・Page 7, Line 234-240. Figure 2 was not described.

The Title of Table 3 and Figure 2, which was sent as a separate file, have been added. Thank you for pointing this out.

Reviewer 2 Report

Comments and Suggestions for Authors

The authors have investigated the role of single-fraction 50Gy carbon-ion radiotherapy for early-stage lung cancer with interstitial pneumonia with regards to its safety and efficacy.

Based on their study, the authors state that CIRT can provide a relatively safe and curative treatment for patients with IP-complicated lung cancer. 

The manuscript is well written and the flow of the manuscript is acceptable. However there some queries that need to be addressed before it can be reconsidered.

Major queries

Results section:

1.when median is given, authors should use interquartile range (IQR)

1.       2. The 2-, and 3/ 5-year LC rates were 82.7% (95% CI: 65.1–92.0%), and 77.8% (95% CI: 57.9– 180 89.2%): this statement is not clear. What does the author want to convey by 3/5- year OS. This needs to be clearly stated in manuscript.

2.       In the univariate and multivariate analysis, it is natural that the lung dosimetry has come out to be positive for IP. However I did not see that the authors have included smoking history and pretreatment lung morbidities in the statistical analysis. Since it is a prospective study, I am sure the data will be there with the PI. I would suggest them to include it in the Univariate and multivariate analysis.

3.       The authors have mentioned that they have used fractionated RT in central tumours. Have they done any comparison between single fraction vs fractionated CIRT. If not they should include it in the results section

Discussion section

1.       The authors have shown that carbon ion therapy can be given in pts with lung cancer having localised IP. However, scientifically for robustness, the authors should discuss and compare with photon DIBH technique and PBT. This comparison will bring out the essence of carbon ion therapy.

2.       Comparison between single fraction and fractionated CIRT should be discussed in the discussion section.

Minor queries

Abstract

1.     Authors should use superior instead of high tumor control

2.     clinical stage of lung cancer (UICC 7th edition) was 1A:1B: 2A:2B = 32:13:4:1 is not correctly stated. please state it correctly in the abstract.

Methods

3.     Typographical errors for eg, spinal in line no103 of the methods section

4.     The authors need to mention from where they had used these dose constraints with the relavent literature study.

Results

4.       In the table-1, next to Patient Characteristics, the authors should mention number of patients.

Conclusion

The authors mention that careful individualized assessments are required to determine the indications for treatment. The authors should be more specific about this statement.  The authors should mention about the indications of such a treatment given that it is expensive and is available in limited centres across the world.

Author Response

Response to Reviewer2 Comments

Thank you very much for taking the time to review this manuscript. Please find the detailed responses below and the corresponding revisions/corrections highlighted/in track changes in the resubmitted files. 

Author's Reply to the Review Report (Reviewer 2)

Major queries

Results section:

  1. when median is given, authors should use interquartile range (IQR)

Thank you for your comments. The description of observation period and patient/treatment characteristics has been adapted from range to IQR.

  1. The 2-, and 3/ 5-year LC rates were 82.7% (95% CI: 65.1–92.0%), and 77.8% (95% CI: 57.9– 180 89.2%): this statement is not clear. What does the author want to convey by 3/5- year OS. This needs to be clearly stated in manuscript.

Thank you for the opportunity for important discussion. IP has a strong impact on survival but are also known to reduce local control of radiotherapy. A brief discussion on local control has been added and the description around it has been revised.

4.4. Treatment effect

…In the present study, mortality from other causes was more frequent than that from cancers, with 2- and 3-year OS of 60.8 and 45%, respectively, which were considerably inferior to the results of single irradiation for early-stage LC without IPF. This is a common trend in reports on radiotherapy for IP-associated LC. Hashimoto et al. and Noh et al. reported 2-year OS of 45% and 71.1%, respectively, which were much worse than that for early-stage LC [40,41]. Kim et al. reported OS after definitive irradiation for stage 1–2 NSCLC, with the IPF group having a four-fold increased risk of death compared to the control group [38]. We should fully consider the prognosis of IP, negative effects of treatment, and patient preferences, and carefully evaluate the appropriateness of interventions, including CIRT. Subsequently, treatment should be conducted as safely as possible.

In the present study, mortality from other causes was higher than that from cancer. The 3-year OS was 45%, which was significantly inferior to our results with the same protocol for non-IP cases [19]. This is a common trend in reports on radiotherapy for IP-complicated LC. Among Japanese multicenter retrospective studies, the study on SBRT re-ported their 3-year OS of 42.6% and that on CIRT 48.2% [33,39]. Other studies have similarly reported relatively poor OS after treatment of IP-complicated LC, regardless of radiation modality [38,40,41]. Meanwhile, it is clear from several reports that the local control of radiotherapy is also reduced in IP-complicated cases [26,40,42,43]. The underlying mechanisms remain unclear, but it has been reported that chronic inflammation and fibrosis could be related to poor tumor response [59]. Also, the difficulty in distinguishing the tumor from the surrounding interstitial shadows could have led to inaccurate targeting and post-treatment assessment, or that efforts to reduce lung irradiation might be negatively reflected. In patients with IP complications, many factors should be fully considered, including the prognosis of IP, the negative effects of the treatment, and the patient's preferences, then a careful assessment should be made of the feasibility of the treatment intervention.

  1. In the univariate and multivariate analysis, it is natural that the lung dosimetry has come out to be positive for IP. However, I did not see that the authors have included smoking history and pretreatment lung morbidities in the statistical analysis.Since it is a prospective study, I am sure the data will be there with the PI. I would suggest them to include it in the Univariate and multivariate analysis.

Thank you for your suggestions. We have added a univariate analysis of smoking history and grade 2-5 pneumonia to Table 5 (also for age and gender). As for pre-treatment lung morbidities, it is represented by respiratory function and history of acute exacerbations. In any case, the number of cases is small and significant results are not available.

  1. The authors have mentioned that they have used fractionated RT in central tumours.Have they done any comparison between single fraction vs fractionated CIRT. If not they should include it in the results section

Thank you for pointing it out. We have treated central lung tumors with a different protocol with a larger number of fractions (68.4Gy/12fr), the same as for SBRT, so they are not included in the current analysis. (We will shortly report on central CIRT in another paper.) From your suggestion, we thought it would be better to state this more clearly, so we have revised part of the '2.1. Study design and patient selection' section. In addition, we have discussed this point together with your comments in the discussion section and revised several descriptions.

2.1. Study design and patient selection (revised 79-81)

Central tumors were not included in this analysis as they have been treated using a different protocol with 12 fractions due to safety concerns.

Discussion section

  1. The authors have shown that carbon ion therapy can be given in pts with lung cancer having localised IP. However, scientifically for robustness, the authors should discuss and compare with photon DIBH technique and PBT. This comparison will bring out the essence of carbon ion therapy.

We have added a few details on the relationship between the Bragg peak (spread-out Bragg peak) and the reduction in the lung low-dose area, as far as we understand at the moment.

4.3. Importance of lung doses. (revised 306-311)

Particle therapy, especially CIRT has excellent depth-dose characteristics known as Bragg peaks. It allows to create a region called the spread-out Bragg peak (SOBP) adapted to the depth and shape of the treatment target with only several beam angles, minimizing dose scattering to the nearby OAR and increasing the dose to the target [13]. As a result, CIRT enables reduced lung doses, particularly in the low dose area, which is probably related to lower toxicity in CIRT for IP-complicated LCs [10,32-34,37].

  1. Comparison between single fraction and fractionated CIRT should be discussed in the discussion section.

Since the establishment of CIRT for early-stage lung cancer, we have treated almost all early-stage lung cancers with single irradiation, and at present it is difficult to make a direct comparison with fractionated irradiation. We have added this point to the limitation section and also stated in the methods section that central tumors are not included in this study.

4.5. Limitation

… Fourth, the safety of single irradiation, which we use in our hospital, has not been established for other modalities at present, making pure inter-modality comparisons difficult. Also, methods to compare the biological effects of CIRT and photon SBRT have not yet been established. Further studies with large sample sizes and basic research are needed to clarify the risk factors of severe RP and the role of CIRT in IP complicated LCs.

Minor queries

Abstract

  1. Authors should use superior instead of high tumor control

Thank you for pointing it out. I have corrected.

  1. clinical stage of lung cancer (UICC 7th edition) was 1A:1B: 2A:2B = 32:13:4:1 is not correctly stated. please state it correctly in the abstract.

Does your point depend on the mixed use of the UICC 8th edition in Table 1? We have reviewed the staging descriptions and have standardized the description of the clinical stage of lung cancer to the UICC 7th edition, including the table, to avoid confusion for the readers. Thank you for your important remarks.

Methods

  1. Typographical errors for eg,spinal in line no103 of the methods section

We are very sorry about the typing error. We have corrected them as far as possible.

  1. The authors need to mention from where they had used these dose constraints with the relavent literature study.

We have added the points you mentioned. Thank you.

2.2. Carbon-ion radiotherapy procedure (revised 104-105)

The total dose was applied to the isocenter and tuned to cover the PTV with a 95% isodose line of the prescribed dose. For organs at risk (OARs), dose constraints were set based on our previous clinical trials and strictly adhered to: spine (Dmax) <10 Gy,…

Results

  1. In the table-1, next to Patient Characteristics, the authors should mention number of patients.

Thank you for pointing this out. I have added.

Conclusion

  1. The authors mention that careful individualized assessments are required to determine the indications for treatment. The authors should be more specific about this statement.  The authors should mention about the indications of such a treatment given that it is expensive and is available in limited centres across the world.

Thank you for your comment. As you mentioned, we thought that the issues of ubiquity and cost of treatment should be mentioned. On the other hand, the superiority of CIRT is still far from proven, and we did not think we could strongly say that this treatment should be widely used as a conclusion of this study. We added a sentence to the limitation section and also improved the expression in the conclusion section.

Limitation (revised 345-346)

This study had some limitations. First, the study was conducted at a single institution and had a retrospective design. In addition, CIRT is not an accessible treatment due to its ubiquity and high treatment costs. …

Conclusion (revised 363-365)

However, IP complications increase the risk of serious RP, and severe cases have been re-ported. Further studies and careful individualized assessments are required to determine the indications for treatment. 

 →

However, IP complications increase the risk of serious RP, as reported in the present study, are certainly high risk even with CIRT. Further research is needed to identify the most beneficial targets and the limitations of this treatment.

Round 2

Reviewer 1 Report

Comments and Suggestions for Authors

This manuscript was correctly modified. I consider that the manuscript is worthy of being accepted.

Reviewer 2 Report

Comments and Suggestions for Authors

The authors have responded to all my queries 

The manuscript can be accepted for publication with minor endlish editing

Comments on the Quality of English Language

Minor english editing is required before acceptance